# Nanostructured Cu_2_O Synthesized via Bipolar Electrochemistry

**DOI:** 10.3390/nano9121781

**Published:** 2019-12-15

**Authors:** Steven McWilliams, Connor D. Flynn, Jennifer McWilliams, Donna C. Arnold, Ruri Agung Wahyuono, Andreas Undisz, Markus Rettenmayr, Anna Ignaszak

**Affiliations:** 1Department of Chemistry, University of New Brunswick, Fredericton, NB E3B 5A3, Canada; Steven.McWilliams@unb.ca (S.M.); connor.flynn@unb.ca (C.D.F.); 2Department of Psychology, University of New Brunswick, Fredericton, NB E3B 5A3, Canada; Jennifer.Sanford@unb.ca; 3School of Physical Sciences, University of Kent, Canterbury CT2 7NH, UK; d.c.arnold@kent.ac.uk; 4Institute for Physical Chemistry and Abbe Center of Photonics, Friedrich-Schiller-Universität, 07743 Jena, Germany; ruri.wahyuono@uni-jena.de; 5Otto Schott Institute of Materials Research, Chair of Metallic Materials, Friedrich-Schiller-Universität, 07743 Jena, Germany; Andreas.Undisz@uni-jena.de (A.U.); M.Rettenmayr@uni-jena.de (M.R.)

**Keywords:** bipolar electrochemistry, green synthesis, substructure, photocurrent, semiconductors

## Abstract

Cuprous oxide (Cu_2_O) was synthesized for the first time via an open bipolar electrochemistry (BPE) approach and characterized in parallel with the commercially available material. As compared to the reference, Cu_2_O formed through a BPE reaction demonstrated a decrease in particle size; an increase in photocurrent; more efficient light scavenging; and structure-correlated changes in the flat band potential and charge carrier concentration. More importantly, as-synthesized oxides were all phase-pure, defect-free, and had an average crystallite size of 20 nm. Ultimately, this study demonstrates the impact of reaction conditions (e.g., applied potential, reaction time) on structure, morphology, surface chemistry, and photo-electrochemical activity of semiconducting oxides, and at the same time, the ability to maintain a green synthetic protocol and potentially create a scalable product. In the proposed BPE synthesis, we introduced a common food supplement (potassium gluconate) as a reducing and complexing agent, and as an electrolyte, allowing us to replace the more harmful reactants that are conventionally used in Cu_2_O production. In addition, in the BPE process very corrosive reactants, such as hydroxides and metal precursors (required for synthesis of oxides), are generated in situ in stoichiometric quantity, providing an alternative methodology to generate various nanostructured materials in high yields under mild conditions.

## 1. Introduction

In recent years, cuprous oxide (Cu_2_O) has become increasingly popular as a semiconductor given its dual ability to convert solar energy and facilitate the photoelectrochemical splitting of water in photovoltaic devices [1,2]. While promising as a semiconductor, the major shortcoming that restricts the application of Cu_2_O in photovoltaics is its photochemical instability [3,4]. Numerous studies have been devoted to improving the stability of Cu_2_O electrodes in solution through surface modification with conducting polymers, metals, and oxides [5,6,7]. These protective layers not only inhibit photo-corrosion of Cu_2_O, but also facilitate the band structure for improved charge transport [6]. Despite the highlighted issue, Cu_2_O remains a thoroughly researched semiconductor material due mainly to the distinct qualities and characteristics that set it apart. Cu_2_O has a narrow band gap (1.8–2.5 eV) as compared to other semiconductor oxides [7], a high abundancy in the earth’s crust, and can be synthesized via a variety of processes that are both low-cost and scalable. When dealing with nanostructured Cu_2_O, the electrochemical and chemical stability, band gap level, flat band potential, and even the nature of semi-conductivity can vary considerably depending on the synthesis method used to produce them [8,9]. The ability to precisely control these qualities of Cu_2_O leads to a number of new and exciting applications. One of the most extensively investigated applications is the Cu_2_O-catalysed, light-driven purification of wastewater [10]. For example, various combinations of metal-Cu_2_O nanoparticles have been used for the decontamination of water polluted with dyes or hydrogen peroxide [11]. Here, Cu_2_O particles are labeled cleaning “nano-swimmers” or “micromotors” as they exhibit self-propelled movement, driven by gases that are generated at one end of the Cu_2_O particle. With this Cu_2_O-based decontamination method, new cleaning suspensions represent a less toxic alternative when compared to conventional water purification methods that produce enormous quantities of secondary waste (e.g., chromium, iron compounds).

Another important application of Cu_2_O is to catalyze organic coupling reactions, including carbon–carbon [12], and nitrogen–carbon couplings [13]. Cu_2_O-catalyzed synthesis (with and without illumination) of amides via amidation of aryl halides [14], Sonogashira- [15], and Suzuki-type couplings [16,17], have shown excellent conversion with satisfactory yields under mild reaction conditions at much lower cost (as compared to palladium-catalyzed syntheses). Moreover, a recyclable Cu_2_O nano-catalyst has been utilized for nitrogen–carbon cross coupling of aryl halides with aromatic/aliphatic amides [18] and amines [13], as well as numerous types of Cu_2_O-driven cycloaddition reactions (e.g., click chemistry [19], Huisgen cycloaddition [20]). The good selectivity and low toxicity, together with excellent yields of the Cu_2_O-catalyzed methods, make this an inexpensive catalyst that is especially important for up-scaled production.

Likewise, Cu_2_O in combination with other oxides or metals has been reported as a very promising catalyst in the light-driven conversion of CO_2_ to fuels and feedstocks [21,22]. For example, Cu-Cu_2_O and Cu-Cu_2_O-ZnO have shown excellent activity towards electrochemical reduction of CO_2_ to higher alcohols, with the Cu-Cu_2_O-ZnO catalyst being the most selective for methanol generation [23]. With sustainable solar energy as a resource, photocatalytic reduction of CO_2_ is in high demand for the production of technologically important gaseous and liquid chemicals, such as methane and methanol. More importantly, Cu_2_O-catalyzed conversion of CO_2_ has enormous potential for use in the reduction of greenhouse gases, which may assist in minimizing one source of climate change.

In previous studies, Cu_2_O has been synthesized through various chemical [24,25], electrochemical [26,27], and mechanochemical pathways [28,29]. Although these methods resulted in Cu_2_O with new structures and photocatalytic efficiencies, many of them pose potential environmental risks as they require hazardous reactants (e.g., sodium hydroxide) or metal precursors (e.g., copper nitrite, copper chloride, copper sulfate), and often generate a high volume of by-product [30]. Thus, a synthesis method that is capable of producing mass amounts of Cu_2_O in a fashion that is both environmentally friendly and relatively inexpensive is of great interest to researchers. One of the most promising techniques in this regard is electrochemical synthesis. This is due to its ability to generate copper ions directly from metals, without the use of additional chemicals. Other benefits of Cu_2_O electro-synthesis include the ability to control the rate of reaction, the simplicity of the setup, the ease with which the reaction can be scaled up, and the generation of highly porous Cu_2_O nanostructures [31]. As an example, Cu_2_O was previously synthesized using a closed bipolar electrochemical setup that utilized four electrodes and CuSO_4_ as the source of copper ions, and was pioneered by Ugo et al. in 2016 [32]. This work demonstrated that porous Cu_2_O deposit can be synthesized simultaneously with metal or metal oxides in one electrochemical reactor, and was used to generate unique structures called Janus particles.

One electrochemical method that has never been utilized in the large-scale production of Cu_2_O is a wireless electro-synthesis process known as open bipolar electrochemistry (BPE).

BPE is a contactless method and consists of a conductive object (BPE electrode) that is placed between two driving electrodes. This BPE electrode develops opposite charges at each end, hence the name bipolar, and can thus facilitate oxidation and reduction reactions simultaneously [33]. Redox reactions in solution are driven forward by overpotentials that form on the extremities of the wireless electrode. The potential difference between two redox reactions occurring at the bipolar electrode, ∆*E*_elec_, is determined by the equation:(1)$$\Delta E_{\mathrm{elec}}=E_{\mathrm{tot}}\left(\frac{l_{\mathrm{elec}}}{l_{\mathrm{channel}}}\right)$$
where *E*_tot_ represents the total applied potential; *l*_channel_ is the distance between the two driving electrodes; and *l*_elec_ is the thickness of the bipolar electrode. When a sufficient potential is applied to the cell, oxidation of the copper plate, paired with the reduction of water, takes place on the wireless and driving electrodes, respectively (Appendix A). In the proposed BPE synthesis, the formation of Cu_2_O species is only possible in the presence of potassium gluconate, which has multiple roles: as an electrolyte (to increase the conductivity of the solution); as a complexing agent (C_12_H_22_CuO_14_); and, most importantly, as a reducing agent, allowing for the generation of Cu^+^ ions. Potassium gluconate is a low-cost food supplement and assists several functions of the body (e.g., it is used to regulate potassium levels in the blood [34]). Herein, we exploit the gluconate as an essential reactant in the synthesis of nanostructured Cu_2_O. This allows for the removal of more harmful reducing agents required in other Cu_2_O syntheses. In addition, one of the most important benefits of BPE is the ability to exclude corrosive reactants, such as hydroxide, that are used in conventional Cu_2_O fabrication. In the BPE process, the hydroxide species are generated in situ via water electrolysis occurring at the driving electrodes (Equation (S2), Appendix A). The formation of Cu^2+^ on the bipolar electrode, accompanied by the production of stoichiometric amounts of OH^−^ on the driving electrode, is controlled by a low applied potential so that no excess of corrosive by-product is generated. In this work, a series of nanostructured Cu_2_O have been synthesized in the BPE reactor. The effect of applied voltage and reaction time on structure and photo-electrochemical characteristics are explored and compared to commercial Cu_2_O. The BPE approach is highly scalable, easy to control, and can open new avenues in the industrial production of various nanostructured materials, as it is both cost effective and safer for the environment.

## 2. Materials and Methods

### 2.1. Materials

Potassium D-gluconate (≥99%, CAS# 299-27-4), fluorine-doped tin oxide (FTO) plates (300 × 399 × 2.2 mm resistivity 7 Ω/cm^2^), and cuprous oxide (≥99.99%, trace metal basis, CAS# 1317-39-1) were purchased from Sigma-Aldrich (Oakville, ON, Canada). Anhydrous ethanol (100%, CAS# 64-17-5) was obtained from Commercial Alcohols and sodium sulfate (>99%, CAS# 7757-82-6) was obtained from Fisher Scientific (Nepean, ON, Canada).

### 2.2. Characterization

#### 2.2.1. X-Ray Diffraction (XRD)

The composition and crystalline structure of cuprous oxide samples was investigated using XRD. All XRD analyses were performed using a Bruker AXS D8 microdiffractometer (Bruker Corp., Billerica, MA, USA) with a Cu-Kα X-ray tube, a wavelength (λ) of 1.54 Å, and an accelerating voltage of 40 kV. All samples were prepared in a circular well and levelled off to ensure a smooth surface for analysis.

#### 2.2.2. X-Ray Photoelectron Spectroscopy (XPS)

XPS analysis was performed to measure the elemental composition of the prepared and commercial cuprous oxide samples. XPS measurements were obtained using a VG Microtech MultiLab ESCA 2000 spectrometer (VG Microtech Limited, London, UK) with 100 μm analyzer spatial resolution, 10 meV energy resolution, and Mg Kα linewidth.

#### 2.2.3. Transmission Electron/Scanning Electron Microscopy (TEM/SEM)

The morphology of cuprous oxide samples was investigated using TEM and SEM at Friedrich-Schiller University (FSU) and the University of New Brunswick (UNB). A suspension of Cu_2_O (0.5 mg/mL) was deposited onto a nickel grid (TEM) or ultra-smooth glassy background (SEM). A JEOL JSM-6400 SEM (Jeol Ltd., Tokyo, Japan), equipped with an EDAX Genesis 4000 energy dispersive X-Ray (EDX) analyzer, was used at an accelerating voltage of 15 kV (UNB). High resolution TEM (HRTEM) was carried out with a JEOL 3010 and JEOL NeoARM TEM (Jeol Ltd., Tokyo, Japan).

#### 2.2.4. UV–Vis Diffusive Reflectance

UV–Vis diffusive reflectance studies were performed using a Lambda 25 UV/VIS spectrometer double beam (DB) instrument (PerkinElmer Inc., Waltham, MA, USA) in the wavelength range from 200 to 1200 nm.

#### 2.2.5. Photoluminescence

Emission measurements were carried out using thin film Cu_2_O. For thin film preparation, a paste of 1 mg/1 mL (MeOH) was drop-cast on a 2 × 1 cm masked glass substrate and dried at 90 °C for 1 h (low temperature drying was used to avoid microstrutcure alteration).

The apparatus used was a FLS980 spectrometer (Edinburgh Instruments, 2 Bain Square Kirkton Campus, UK). A NIR PMT detector was used for wavelength scans beyond 850 nm.

#### 2.2.6. Electrochemical Characterization

All electrochemical experiments were conducted using a three-electrode system with a Cu_2_O|FTO plate working electrode, a platinum wire counter electrode, and an Ag/AgCl (saturated KCl, *E* = 0.197 V) reference electrode in 0.5 M sodium sulfate. A CHI 660-E potentiostat (CH Instruments Inc., USA) was used for all electrochemical tests. Photocurrent was analyzed under open circuit potential (OCP) by subjecting the working electrode to intermittent illumination in fifty second intervals using an 85 W Xe Sunworld HID light source. Mott–Schottky (MS) analysis was performed under a reverse bias from 0.1 to 0.6 V at 10 kHz, with a potential step of 0.05 V. The voltage range for MS and photocurrent tests was determined by cyclic voltammetry (CV). The potential range with no copper redox activity was used for MS analysis. AC impedance was carried out, without applied bias, under dark and illuminated conditions and in a frequency range of 10^−2^–10^5^ Hz.

#### 2.2.7. Fabrication of Cu_2_O Electrode

Prior to the casting of Cu_2_O, FTO glass plates were cleaned in acetone and ethanol by ultrasonication, and then washed with deionized (DI) water. 10 mg of Cu_2_O in 5 mL of anhydrous ethanol was sonicated for 1 h. Finally, 125 μL of Cu_2_O suspension was deposited onto the FTO plate and allowed to dry at ambient conditions (the area of electrode was 1 cm^2^ for all samples).

#### 2.2.8. Synthesis Procedure

Potassium gluconate solution was used as both an electrolyte and complexing agent for the free copper ions released during the BPE process from the Cu plate. 4.5 g of potassium gluconate was dissolved in one litre of pure DI water (19.2 mM) reaching pH = 6.8. 200 mL of the potassium gluconate solution was heated to 80 °C. A copper plate and driving electrodes were secured in a custom-made Teflon top. A constant potential was applied for 1 h at 4.5, 5.0, 6.0, 7.0, and 8.0 V, or for 3 h at 4.5 V, as presented in Appendix A. The precipitate was filtered, washed three times with distilled water and then with ethanol, dried, and stored in ambient conditions. A reaction mixture was sampled at time intervals to demonstrate the various stages of the reaction for the Cu_2_O synthesized at 6.0 V (Appendix A). At 0 min, the solution consisted of a copper plate that was immersed in 19.2 mM potassium gluconate solution. The applied potential resulted in the oxidation of the copper plate (Appendix A, Equation (S1)), which was paired with the reduction of water (Equation (S2)). At 5 min, a dilute concentration of copper gluconate had formed (Equation (S3)), which was demonstrated by the slight blue color of the solution. This converted into a green tint by 10 min due to the generation of copper (I) hydroxide (CuOH). The CuOH formed due to the reduction of the Cu^2+^ by glucose (Equation (S4)) and the subsequent reaction with free hydroxide ions (Equation (S5)). As the concentration of CuOH increased, and Cu_2_O was subsequently generated by degradation of CuOH (Equation (S6)), the solution color changed to yellow over 20 min (Appendix A). Finally, due to the increased concentration and the size of the Cu_2_O particles, the colour of the suspension darkened to orange. All samples exhibited the same steps in the synthesis process; however, the color changes occurred faster as the synthesis potential increased. The reaction steps are shown in Appendix A in both sampled solution (top photos) and in the BPE electrochemical cell (bottom photos).

## 3. Results

### 3.1. Structural Analysis of Cu_2_O

The effects of varying potential and time on the structure of as-synthesized Cu_2_O was investigated using X-ray diffraction (XRD), followed by a Rietveld refinement analysis. Initial observations indicated that all synthesized material was phase-pure due to the absence of metallic copper and CuO. As represented in Figure 1, Cu_2_O samples exhibited the most intense XRD peak at approximately 36.5 2θ, indicating preferred growth towards a 111 orientation. All BPE-generated oxides showed much broader XRD patterns and, thus, had smaller crystallite sizes compared to the commercial Cu_2_O. In addition, referring to the commercial Cu_2_O, a slight shift was observed towards higher values of 2θ for all BPE-synthesized samples (Figure 1a–c). These shifts in BPE-synthesized oxide were assumed to be caused by either stress-strain or structural defects in the lattice.

Both crystallite size and lattice constants are presented in Table 1. The particle size was calculated using Scherrer’s equation [35] (Equation (2)), where *D* is the crystallite size in nm, *λ* is the Cu K*α* radiation wavelength (1.5046 Å), *K* is the shape factor (0.9), *β*_hk*l*_ is the full width at half maximum (FWHM) in radians, and *θ* is the scattering angle in radians.
(2)$$D=\frac{K\lambda }{\beta_{\mathrm{hkl}}\mathrm{Cos}\theta }$$

As seen in Table 1, crystallite size decreased as the applied potential in the BPE synthesis increased, with a minimum (14.5 nm) occurring at the maximum applied potential of 8.0 V. Similarly, by extending the reaction time from 1 to 3 h (sample synthesized at 4.5 V, red and blue patterns in Figure 1), crystallite size increased from 21.7 to 27.1 nm. Thus, using BPE to synthesize Cu_2_O, crystallite size can be decreased by either increasing the applied voltage or decreasing the synthesis duration.

Rietveld analysis (Appendix A) was performed for the diffraction data collected for all materials using the GSAS software [36,37] and the model for Cu_2_O proposed by Foo et al. (as taken from the ICSD) [38]. Refinements were performed for twenty variables, which included lattice parameters, atom positions, zero-point, peak shape, and background. The background was modelled with twelve terms via a shifted Chebyschev polynomial function. The peak shape was modelled using the pseudo-Voigt function, as described by Howard and Thompson et al. [39,40]. To mitigate surface roughness effects that were introduced by preparation of the sample for X-ray diffraction analysis, six spherical harmonic order terms were also refined in a cylindrical geometry. In all cases, the texture index was close to 1, indicating that the sample was randomly oriented. Refinement data is included in Table 1, with the refinement profiles given in Appendix A. All materials showed an excellent fit to the proposed model. The lattice parameters, as listed in Table 1, demonstrated no significant impact on lattice structure or Cu–O bond lengths when either applied potential in BPE synthesis or reaction time were altered. However, as previously discussed, increasing applied potential did allow for a decrease in the overall crystallite size. Therefore, apart from crystallite size, it is assumed that time and potential played no further role in augmenting the presented samples.

Photoluminescence (PL) analysis was carried out at various excitation wavelengths on all Cu_2_O samples. PL data (Appendix A) confirmed an insignificant amount of structural defects in all BPE-synthesized and commercial samples through the lack of oxygen and copper vacancy peaks, which typically appear around 700–750 nm and 910–920 nm, respectively [41]. This apparent absence of structural defects suggests that any shifts in the XRD patterns of as-synthesized Cu_2_O are the result of lattice strain.

### 3.2. Morphology

Figure 2a–d presents SEM images of Cu_2_O synthesized at 4.5 V-1H, 6.0, 7.0, and 8.0 V. Agglomeration for all samples was significant despite several attempts to improve dispersal using surfactants (PVP, SDS, CTAB, Triton X-100, HMT); thus, individual structure determination was impossible. All images presented show samples without added surfactants. SEM observations further demonstrated that Cu_2_O agglomerates were porous and composed of much smaller structures.

HRTEM images of 4.5 V-1H and 7.0 V samples are shown in Figure 2e,f, respectively. The individual size of particles revealed by TEM agree well with the crystallite size calculated from XRD data (Table 1). Crystallites imaged through TEM contain several different substructures (Figure 2, red and blue circles) composed of similarly oriented lattice planes. Substructures in the 4.5 V-1H sample were much smaller than those in the 7.0 V sample, suggesting a higher variability in lattice orientation with lower applied voltages. The green circles in Figure 2e,f highlight areas of changing lattice orientation and contain two or more overlapped substructures.

While the Rietveld analysis of BPE-synthesized Cu_2_O suggested no significant difference in particle shape with varying synthesis times and potentials, TEM/SEM of commercial Cu_2_O revealed considerable differences between the as-synthesized and commercial products (Appendix A). The commercial material showed much larger and more irregular agglomerates that were bigger than 1 μm. The crystallite size for the reference material, calculated by Equation (2), was also much larger (92.9 nm) than oxides produced in this work. Therefore, the proposed bipolar electrosynthesis can produce nano-sized Cu_2_O particles that are both phase-pure and defect-free.

### 3.3. X-Ray Photoelectron Spectroscopy (XPS): Surface Chemistry

Figure 3a shows an example of the O 1s signal for BPE-made Cu_2_O (6.0 V). Signal deconvolution reveals two main components: the peak at 531.4 eV represents the O–Cu(+1) in Cu_2_O, and the larger peak at 532.0 eV generally represents Cu_2_O, Cu(OH)_2_, and/or CuO. Since deconvolution of the Cu 2p_3/2_ signal (Figure 3b) identified Cu(+1) as the main signal, the O 1s peak at 532.0 eV represents either a defective oxygen lattice in Cu_2_O (not identified in the BPE-synthesized Cu_2_O based on photoemission analysis, Appendix A), or the presence of adsorbed oxygen from water or carbonates [42]. As shown in Figure 3b, the main Cu_2_O peak was observed at 932.4 eV, with a weak Cu(II) signal appearing at 935.2 eV. This weak Cu(II) peak may be the result of minor surface oxidation due to long-term air exposure or the presence of Cu(OH)_2_ residue formed during the synthesis [40]. Regardless of this impurity, XRD analysis of product stored in ambient conditions for an extended time identified Cu_2_O as the only phase. Furthermore, the Cu 2p_3/2_ signal (Figure 3b) confirmed the absence of both metallic copper and CuO, validating that the material was phase-pure. The Cu LMM peak identified at 952.3 eV (Figure 3b) corresponds to Cu_2_O [43,44]. Figure 3c compares the O 1s signals for all Cu_2_O samples. Signals for as-synthesized oxides are slightly shifted towards lower binding energies, with the 4.5 V-3H sample peak nearly matching the commercial sample.

Despite shifts in the Cu 2p_3/2_ peaks (Figure 3d), no trend related to the applied synthesis voltage was observed. These chemical shifts could be related to several factors, including lattice strain, surface oxygen defects, or differences in particle size. An important observation is that Cu_2_O generated at 8.0 V showed no signs of degradation through the absence of Cu 2p_3/2_ satellite peaks (~943.0 eV), unlike the 6.0 and 7.0 V samples (Figure 3d).

### 3.4. Optical Studies: Diffusive-Reflectance UV–Vis and Kubelka–Munk Analysis

The diffusive reflectance spectra of the commercial and BPE-synthesized samples are presented in Figure 4a. As-prepared oxides showed better light absorption than the commercial material in the range from 200–420 nm. This increase in absorption is best explained by the higher concentration per unit area (higher surface area) of nanometer-sized oxides made in this work when compared to the reference material. Cu_2_O prepared by the BPE method demonstrated a gradual increase in reflectance, starting at approximately 450 nm (all samples). In contrast, the % reflectance of the commercial Cu_2_O showed a sharp increase around 600 nm. This difference can be explained by the effect of particle size—decreasing the particle size induced an increase in the band gap of the material, which altered the ideal absorption wavelength (i.e., blue shifting nanomaterial). The band gap energies were determined by applying the Kubelka–Munk function:(3)$$F\left(R\right)=\frac{K}{S}=\frac{1-R_{\infty }}{2R_{\infty }}$$
where *R* is the absolute value of reflectance, *F(R)* is the absorption coefficient equivalent, *K* is the absorption coefficient, *S* is the scattering coefficient, and *R_∞_* is the diffusive reflectance of the film. A Tauc plot (Figure 4b) correlates the Kubelka–Munk function and energy of a photon, *(F(R)* × *E*_photon_*)^2^* = f(*E*_photon_), enabling extrapolation of the band gap energy. For the commercial material, the band gap energy is close to values reported in the literature (~2 eV) [45]. Interestingly, Cu_2_O prepared via BPE exhibited a higher band gap energy of approximately 2.5 eV for all samples. This increase in energy is caused by the quantum confinement effect, which is the generation of discrete energy levels in the valance and conduction bands. This, in turn, results in an increase in the band gap and is often observed in nano-sized materials [46,47]. The optical analysis revealed two competing effects: Higher band gap energies in BPE-made Cu_2_O due to its smaller particle size (possibly due to the presence of substructures), and an increased light absorption below 450 nm, as compared to the reference material. This demonstrates that the nano-sized Cu_2_O prepared by BPE may scavenge light more efficiently when compared to the commercial sample.

### 3.5. Mott–Schottky Analysis: Flat-Band Potential and Carrier Concentration

Mott–Schottky (MS) analysis resulted in estimation of the flat band potential (*E*_fb_), charge carrier concentration, and type of semi-conductivity [48]. The space-charge capacitance was determined from the imaginary section of the impedance. MS plots are presented in Figure 5a for all samples, including the commercial oxide. Both the reference and synthesized Cu_2_O demonstrated a negative slope for the MS function, which indicated a p-type semi-conductivity. For p-type semiconductors, the majority of carriers are electron holes and the carrier concentration can be determined by Equation (4).
(4)$$\frac{1}{C^{2}}=\left(\frac{2}{\varepsilon \epsilon_{0}*{N_{a}}^{2}}\right)\left(V-V_{\mathrm{fb}}-\frac{kT}{e}\right)$$
where *e* is the elementary charge of an electron, *N*_A_ is the acceptor density, *ϵ* is the dielectric constant of Cu_2_O (7.26), *ϵ*_0_ is the permittivity of free space, *T* is the temperature in K, and *k* is Boltzmann’s constant. The flat band potential (*V*_fb_) and carrier concentration are estimated by extrapolating the linear region of the MS function to determine the x-intercept and utilizing the slope in Equation (5), respectively.
(5)$$\mathrm{Slope}=\left(\frac{2}{\epsilon \epsilon_{0}{N_{a}}^{2}}\right)$$

The flat band potential and carrier concentration are shown in Figure 5b,c. As the flat band potential increased, the carrier concentration increased as well. The effect of applied voltage in the BPE synthesis was observed above 5.0 V. 8.0 V had the largest flat band potential and carrier concentration at −0.11 V and 1.57 × 10^17^ cm^−3^, respectively, which represents an overall increase of 0.136 V across the BPE potential range. The synthesis time for materials synthesized at 4.5 V had almost no influence on carrier density. A slightly lower flat band potential was observed for Cu_2_O made at extended time (3 h). It can be assumed that for the BPE-synthesized Cu_2_O, the differences in both *V*_fb_ and *N*_a_ are strongly related to structural changes and surface chemistry, as demonstrated by XRD and XPS. Moreover, the presence of ultra-fine substructures within the particle—the nature of the grain boundaries—may explain the observed trends in *V*_fb_ and *N*_a_. On the other hand, the commercial oxide had a higher *V*_fb_ and *N*_a_ than all Cu_2_O synthesized in this work. This effect is caused by the increased crystallinity of the reference Cu_2_O as compared to particles on the nanoscale.

### 3.6. Photo-Electrochemistry: Carrier Lifetime and Photocurrent

Analysis of Bode-phase plots (Figure 6a) was exploited to calculate the charge recombination time in dark and illuminated conditions. By evaluating the maximum peak frequency, the recombination times can be calculated using Equation (6).
(6)$$\tau =\frac{1}{2\pi f_{max}}$$
where *τ* is the carrier lifetime and *f*_max_ is the maximum frequency. Figure 6b depicts the calculated carrier lifetimes, with 4.5 V-3H having the longest lifetime at 38 ms. Significantly shorter lifetimes (two orders of magnitude shorter) were observed for Cu_2_O made at higher voltages. Comparatively, the reference sample showed a carrier lifetime within the range of those synthesized by BPE, except for samples made at 4.5 V. The significant increase in carrier lifetime for the 4.5 V-3H is beneficial for photocatalysis due to a reduced rate of electron-hole recombination that allows electrons to participate in photocurrent generation. In total, this analysis led to the conclusion that the charge recombination rate varied for samples synthesized at different times (4.5 V-1H and 4.5 V-3H), but was not influenced by the applied synthesis voltage.

Photocurrent density was determined using an *I-t* (current-time) curve under open circuit potential (OCP) and intermittent illumination. As demonstrated in Figure 7, for Cu_2_O synthesized at higher potentials and longer reaction times, an increase in photocurrent was observed. This is likely due to the difference in structure/morphology (in particular, the particle size effect and the presence of ultra-fine substructures), resulting in various grain boundaries [49], as demonstrated by TEM imaging. The trend observed can be further correlated with the increase in carrier concentration and flat band potential, as demonstrated in Figure 5b,c. The higher carrier concentration and flat band potential are related to smaller particle size (for samples that were synthesized at higher BPE voltage). These two factors allowed for improved total current generation by introducing greater band bending. This contributed to a better charge separation, while simultaneously increasing the concentration of separated charges [48]. Furthermore, the increase in the carrier lifetime for Cu_2_O synthesized at 4.5 V-3H allowed for a larger photocurrent due to better transfer of the carriers. This increase was expected due to better crystallinity, which is closer to that of the commercial Cu_2_O. However, as indicated by the initial sharp peak (Figure 7), transient current peaks appeared upon illumination across all samples that were synthesized via BPE. Transient peaks occurred due to a large degree of surface electron recombination [50,51]. Carriers that are generated in the space charge region are accelerated towards the semiconductor-electrolyte junction, which builds up until the rate of water redox reactions are balanced with the arrival of the carriers. The large surface recombination then reduces the photocurrent that is generated by Cu_2_O due to inefficient carrier extraction. This effect weakens for Cu_2_O that is synthesized at higher voltages. Presumably, a more efficient charge transfer occurs for smaller particles that have larger surface contact with the electrolyte. This effect can be further influenced by combining oxides with more conductive additives for better photo-carrier harvesting [51]. Lastly, the commercial material demonstrated a significant decrease in photocurrent with respect to the synthesized samples. This effect is likely caused by the decreased photoabsorbance of the material resulting in a negative impact on the photogenerated carriers. Therefore, as the synthesized material has significantly higher photocarrier generation, the photocurrent is expected to increase.

## 4. Conclusions

We demonstrate a wireless bipolar electrochemical synthesis of cuprous oxide, targeting the impact of reaction conditions (applied potential and reaction time) on structure, morphology, surface chemistry, and photo-electrochemical activity. SEM/TEM imaging revealed porous-agglomerated materials composed of ultra-fine substructures (2–5 nm) within a single particle in oxide synthesized at low potentials. UV–Vis studies and Kubelka–Munk analysis demonstrated that there was no significant shift in wavelength absorbance/band gap across the synthesized samples. Referring to commercial material, the increase in the band gap and blue shift in the BPE-made Cu_2_O was related to the decrease in particle size. Mott–Schottky analysis verified a moderate change in both carrier concentration and flat band potential among samples synthesized in the BPE process. The observed higher carrier concentration and flat band potential are related to the smaller particle size for samples that were synthesized at higher BPE voltage. These two factors allowed for improved total current generation.

As compared to the reference material, Cu_2_O that was synthesized through BPE demonstrated an increase in photocurrent, a decrease in particle size, and a decrease in recombination at low potentials. However, a decrease in carrier concentration and flat band potential was observed, which was likely induced by the decrease in particle size and, therefore, crystallinity. Ultimately, this research demonstrates that the ideal conditions for BPE synthesis of Cu_2_O have to be maintained at a high potential and for an extended duration of time in order to increase crystallite size. This could potentially minimize carrier recombination and increase photocurrent generation in BPE-generated oxides. Therefore, the conditions of bipolar synthesis can have a significant impact on Cu_2_O crystal properties, as well as the ability to simultaneously maintain a green synthetic route, exclude hazardous waste products, and potentially create a scalable product. Further research is required in order to increase the photocurrent (e.g., pH, doping, complexing agents); all of which are known to have impacts on Cu_2_O photocatalysis.

## Figures and Tables

**Figure 1 nanomaterials-09-01781-f001:**
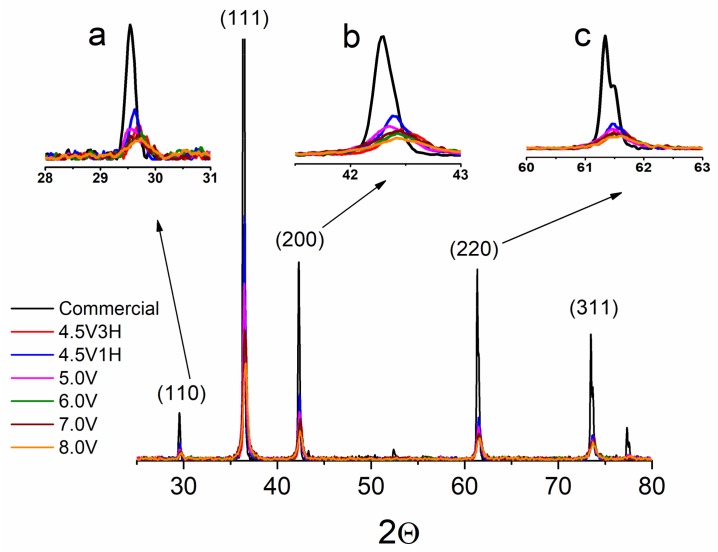
X-ray diffraction patterns of commercial Cu_2_O (black) and Cu_2_O synthesized by bipolar electrochemistry at different applied voltages and synthesis times. The reference signals are assigned to cubic Cu_2_O according to JCPDS (The Joint Committee on Powder Diffraction Standards) card number 00-005-0667. Inserts a, b, and c are enlarged patterns (110), (200) and (220), respectively.

**Figure 2 nanomaterials-09-01781-f002:**
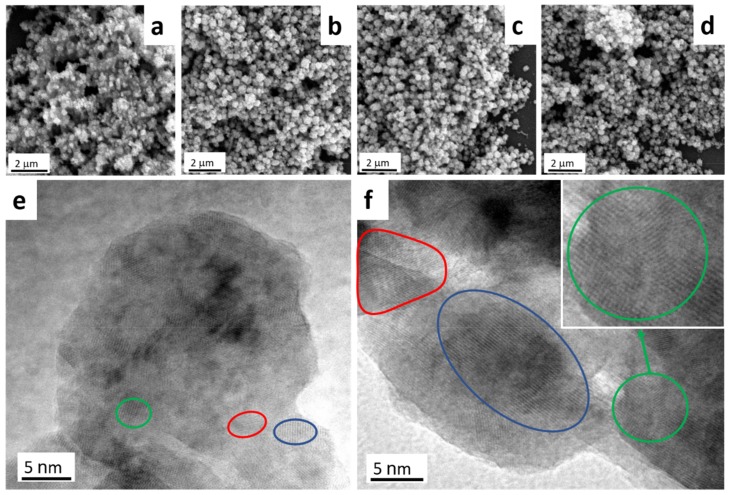
SEM images of 4.5 V-1H, 6.0, 7.0, and 8.0 V Cu_2_O samples (**a**–**d**, respectively) and HRTEM images of 4.5 V-1H (**e**) and 7.0 V (**f**) samples.

**Figure 3 nanomaterials-09-01781-f003:**
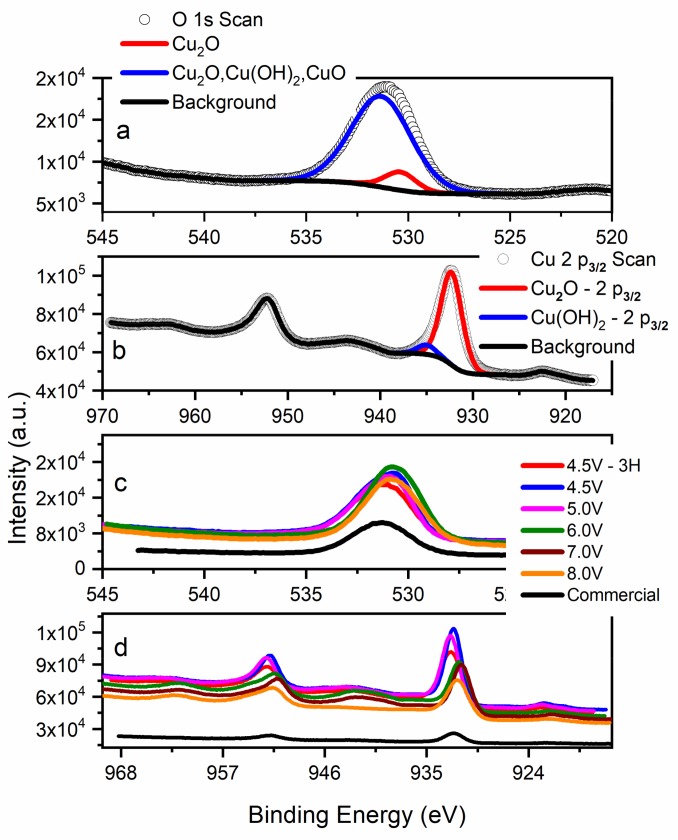
X-ray photoelectron spectroscopy of O 1s (**a**) and Cu 2p (**b**) signals for the 6 V sample; and overlay of O 1s (**c**) and Cu 2p (**d**) spectra for the commercial (black) sample and samples synthesized by bipolar electrochemistry at different applied voltages and times.

**Figure 4 nanomaterials-09-01781-f004:**
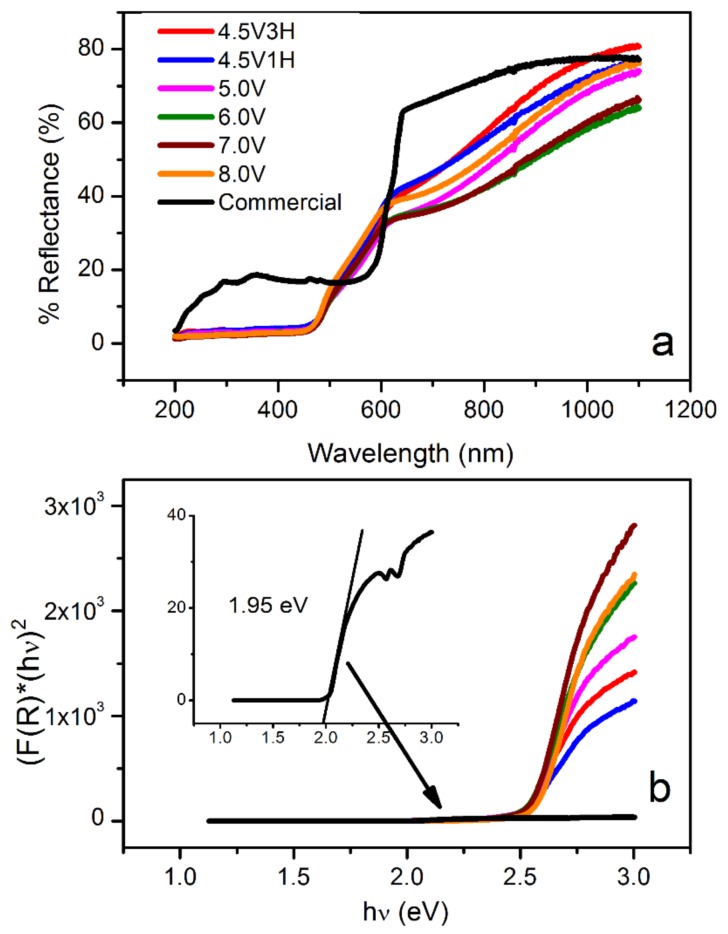
UV–Vis diffusive reflectance spectra (**a**) and Kubelka–Munk function versus bandgap energy (**b**) for commercial Cu_2_O (black) and Cu_2_O synthesized by bipolar electrochemistry at different applied voltages and times.

**Figure 5 nanomaterials-09-01781-f005:**
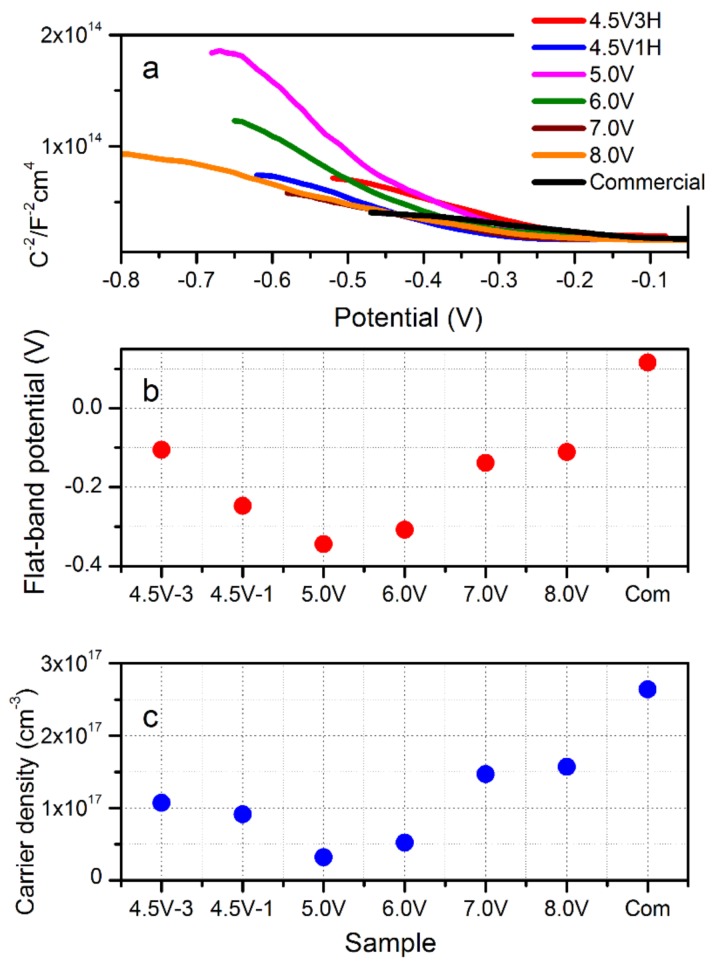
Mott–Schottky (MS) plots (**a**); flat-band potential estimated from linear fit of MS plot (**b**); and carrier density (**c**) calculated from Equations (4) and (5).

**Figure 6 nanomaterials-09-01781-f006:**
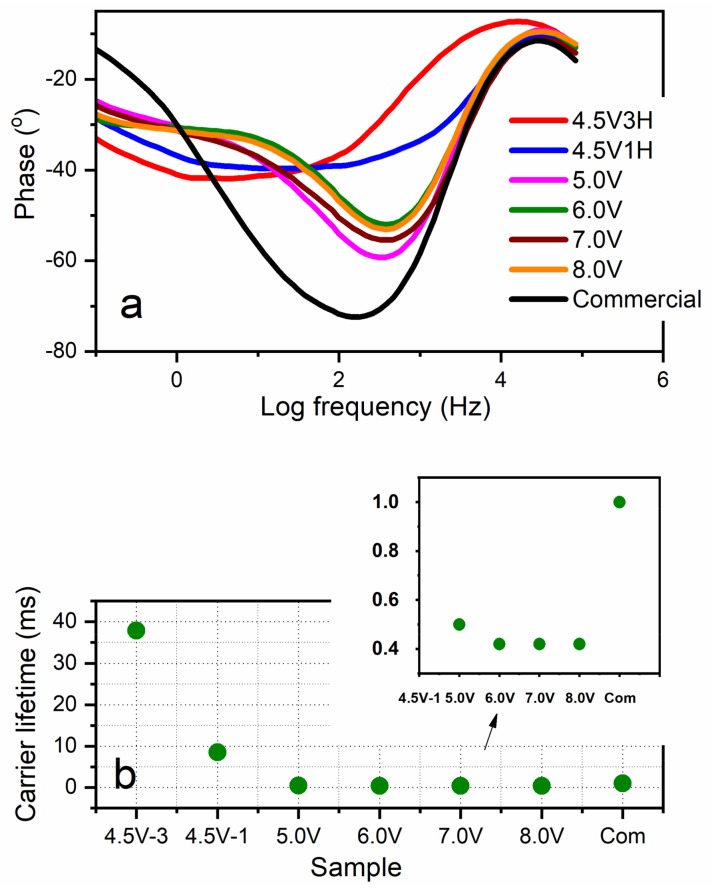
Bode plots for illuminated Cu_2_O electrodes synthesized by bipolar electrochemistry and for commercial Cu_2_O (**a**); carrier lifetime calculated from Equation (6) (**b**).

**Figure 7 nanomaterials-09-01781-f007:**
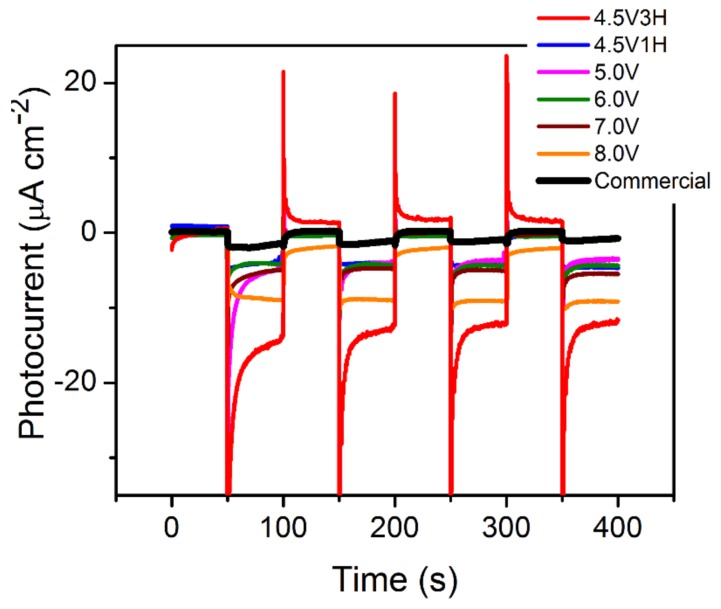
Photo-current density response of Cu_2_O electrodes with illumination intervals of 50 s.

**Table 1 nanomaterials-09-01781-t001:** Refinement data collected from the Rietveld analysis of room temperature XRD data for BPE-synthesized Cu_2_O. All samples showed very good correlation with the Cu_2_O model (space group: *Pn-**3 m*, with atom positions for Cu of (0,0,0) and O of (¼, ¼, ¼)). Parameter *χ^2^* is goodness-of-fit, *R*_wp_ is weighted profile, *R*_p_ is profile residual, and *a* is the lattice parameter in Angstroms (Å). Crystallite size was calculated from the Scherrer formula (Equation (2)).

Parameter	4.5 V-3H	4.5 V-1H	5.0 V	6.0 V	7.0 V	8.0 V
*χ* ^2^	1.17	1.32	1.31	1.33	1.42	1.29
*R*_wp_ (%)	10.84	10.03	9.94	10.46	10.14	10.42
*R*_p_ (%)	8.51	7.95	7.79	8.01	7.93	8.07
*a* (Å)	4.26	4.26	4.27	4.26	4.26	4.26
Cell Vol (Å^3^)	77.53	77.53	77.69	77.36	77.42	77.43
Cu-O bond length (Å)	1.85	1.85	1.85	1.84	1.84	1.84
Cu-O-Cu bond angle (˚)	109.47	109.47	109.47	109.47	109.47	109.47
Crystallite size (nm)	27.1	21.7	20.3	16.2	15.6	14.5

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
