# Peer review of "Nanostructured Cu2O Synthesized via Bipolar Electrochemistry"

_nanomaterials, 2019, doi:10.3390/nano9121781_

Round 1

Reviewer 1 Report

In this research paper, the authors describe the generation of cuprous oxide (CuO2) using bipolar electrochemistry and compare its properties to commercially available CuO2. The article is very well written and provides a very good background into the various methods currently used to generate CuO2 as well as the strategy employed in this study. Importantly, the authors do a very good job explaining the advantages of the bipolar electrochemistry strategy. The materials prepared are very well characterized and the authors describe how changes in the bipolar electrochemistry process alter the properties of the resulting material. The only revision I recommend for this work is for the authors to reduce the number of significant digits used in Table 1 to reflect the limits of XRD measurements (typically 0.01 Å). Therefore, it is the recommendation of this reviewer that the manuscript be accepted after minor revision.

Author Response

Thank you very much for remark. We decreased a number of decimal places in reported value as recommended by Reviewer (in revised version).

Reviewer 2 Report

The manuscript with the ID “nanomaterials-662840” deals with the preparation and characterization of cuprous oxide electrode via an innovative route (bipolar electrochemical approach, BPE).

The authors prepare different types of electrodes by varying the electrolysis potential and time.

The work is very interesting and well described. The electrodes are fully characterized.

I recommend the publication with minor revisions.

Only few considerations:

1) the BPE approach in the preparation of Cu2O was considered in another paper: Closed Bipolar Electrochemistry for the Low-Potential Asymmetrical Functionalization of Micro- and Nanowires, Ugo et al., ChemElectroChem 2016, 3, 450. So, I suggest a revision of the introduction considering this work.

2) line 157: a round bracket is missing after OCP

3) line 184: CuOH instead of Cu2OH

4) line 224: Table 1 instead of Table. 1

5) line354: something is wrong (. Figure 6a.)

6) line 367: the caption refers to a dotted line but there is no graph

Author Response

Response to Reviewer:

1) The BPE approach in the preparation of Cu2O was considered in another paper: Closed Bipolar Electrochemistry for the Low-Potential Asymmetrical Functionalization of Micro- and Nanowires, Ugo et al., ChemElectroChem 2016, 3, 450. So, I suggest a revision of the introduction considering this work.

Response: Yes, this is right – thank you. We initially overlooked this article and added it to the introduction and list of publications in revised manuscript. We also change statement that our bipolar synthesis of Cu2O is first ever reported. The relevant correction appears in the Introduction and conclusions:

In introduction – corrected line 84-88.

As an example, Cu2O was synthesized using a closed bipolar electrochemical setup that utilizes four electrodes and CuSO4 as the source of copper ions, and was pioneered by Ugo et al. in 2016 [32]. This work demonstrated that porous Cu2O deposit can be synthesized simultaneously with metal or metal oxides in one electrochemical reactor, and was used to generate unique structures called Janus particles.

In conclusions – corrected first sentence (line 401-403): We demonstrate a wireless bipolar electrochemical synthesis of cuprous oxide, targeting the impact of reaction conditions (applied potential and reaction time) on structure, morphology, surface chemistry and photo-electrochemical activity.

2) Line 157 (now line 160): a round bracket is missing after OCP

Response: Added

3) Line 184 (now line 186): CuOH instead of Cu2OH

Response: Corrected to CuOH

4) Line 224 (now line 225): Table 1 instead of Table. 1

Response: Corrected to Table 1

5) Line 354 (now line 355): something is wrong (. Figure 6a.)

Response – corrected to: Analysis of Bode-phase plots (Figure 6a) was exploited to calculate the charge recombination time in dark and illuminated conditions.

6) Line 367 (now line 369): the caption refers to a dotted line but there is no graph

Response – corrected to: Bode plots for illuminated Cu2O electrodes synthesized by a bipolar electrochemistry and for commercial Cu2O (a); carrier lifetime calculated from equation 6 (b).

Other Changes:

Added “a” before “BPE” (line 16). Reference 34 was changed to a better reference for the cited information. Removed ‘the’ before “structure-correlated” (line 17). Changed “their average crystallite size was 20 nm” to “had an average crystallite size of 20 nm” (line 19). Deleted “and” after “activity” and added a comma (line 20). Added a comma after “surface chemistry” (line 21). Added a comma after “reactants” (line 26). Added a comma after “(required for synthesis of oxides)” (line 27). Added “that” after “shortcoming” (line 35). Changed “restricting” to “restricts” (line 35). Added “the” before “application” (line 36). Deleted “through” before “click chemistry” (line 62). Deleted “or” before “Huisgen cycloaddition” (line 62). Added a comma after “low toxicity” and after “methods” (line 63). Added “that is” before “especially important for up-scaled production” (line 64). Changed “i.e.” to “e.g.,” before “sodium hydroxide” (line 76). Added “e.g.,” before “copper nitrite” (line 77). Deleted “or” before “copper sulfate” (line 77). Changed the comma after “electrochemical synthesis” to a period (line 80). Added “This is” before “due” (line 80). Added a comma after “metals” (line 81). Changed “are” to “includes” (line 82). Deleted “an” before “open bipolar” (line 90). Changed “i.e.” to “e.g.,” before “it is used to regulate” (line 107). Added a comma before and after “such as hydroxide” (line 111). Deleted “This is in hopes of determining a potential future method that is scalable, cost effective, and environmentally benign.” (lines 115-117). Added “as it is cost effective” after “nanostructured materials” (line 119). Changed “drop casted” to “drop-cast“ (line 151). Changed “a” to “an” before “Ag/AgCl” (line 157). Added a comma after “range” (line 163). Added a comma after “occurred” (line 164). Changed “Furthermore” to “Finally” (line 170). Changed “eq. S1” to “Eq. S1” after “Figure S1” (line 183). Changed “eq. S2” to “Eq. S2” (line 183). Changed “eq. S3” to “Eq. S3” (line 184). Changed “eq. S4” to “Eq. S4” (line 186). Changed “eq. S5” to “Eq. S5” (line 187). Changed “eq. S6” to “Eq. S6” (line 188). Added a comma before and after “thus” (line 201). Delete a space between “1” and “a-c” (line 203). Added a hyphen before “zero” and “point” (line 219). Deleted the comma after “(black)” (line 238). Changed “-“ to “;” after “(PVP, SDS, CTAB, Triton X-100, HMT)” (line 250). Deleted the space between “2” and “e” (line 254). Added a space between “4.5” and “V” in “4.5V-1H” (line 257). Deleted a comma and added “and” after “larger” (line 264). Added an indent at the beginning of the paragraph (line 286). Changed “nanometer-size” to “nanometer-sized” (line 299). Changed “caused” to “explained” (line 302). Added “Where” before “R” (line 307). Added an indent at the beginning of the paragraph (line 307). Added “than” after “efficiently” (line 319). Added an indent at the beginning of the paragraph (line 333). Changed the “s” in “utilising” to a “z” (line 336). Added an indent at the beginning of the paragraph (line 359). Added a comma after “chemistry” (line 403). Deleted the “:” after “demonstrated” (line 413). Changed the “;” to a “,” after “photocurrent” and “size” (line 414). Changed “i.e.,” to “e.g.,” (line 423). Deleted “and” after “doping,” (line 423). Added “and” after “project administration” (line 444). Added a “;” after “D.C.A.” (line 444). Added a comma after “writing” (line 444).